# Antioxidant Activity and GC-MS Profile of Cardamom (*Elettaria cardamomum)* Essential Oil Obtained by a Combined Extraction Method—Instant Controlled Pressure Drop Technology Coupled with Sonication

**DOI:** 10.3390/molecules28031093

**Published:** 2023-01-21

**Authors:** Nora E. Torres Castillo, Giselle D. Teresa-Martínez, Maritza Alonzo-Macías, Carmen Téllez-Pérez, José Rodríguez-Rodríguez, Juan Eduardo Sosa-Hernández, Roberto Parra-Saldívar, Elda M. Melchor-Martínez, Anaberta Cardador-Martínez

**Affiliations:** 1School of Engineering and Science, Tecnologico de Monterrey, Monterrey 64849, Mexico; 2School of Engineering and Sciences, Tecnologico de Monterrey, Queretaro 76130, Mexico; 3Institute of Advanced Materials for Sustainable Manufacturing, Tecnologico de Monterrey, Monterrey 64849, Mexico

**Keywords:** *Elettaria cardamomum*, essential oil, extraction, antioxidant activity, volatile compounds, scanning electron microscopy

## Abstract

Cardamom Essential oils are highly demanded because of their antimicrobial, anti-inflammatory, and antioxidant activities. Nonetheless, retrieving quality extracts quickly with efficient energy savings has been challenging. Therefore, green technologies are emerging as possible alternatives. Thus, this study evaluates the yield and quality of the instant controlled pressure drop (DIC) process coupled with ultrasound-assisted extraction (UAE) of cardamom essential oil (CEO). Likewise, the antioxidant activity, chemical profile of CEO, and microstructure of seeds were analyzed. This study analyzed 13 different treatments with varying saturated steam processing temperatures (SSPT), thermal processing times (TPT), and 1 control. The results showed that CEO yield increased significantly by DIC (140 °C and 30 s) and UAE compared to the control (22.53% vs. 15.6%). DIC 2 (165 °C, 30 s) showed the highest DPPH inhibition (79.48%) and the best Trolox equivalent antioxidant capacity (TEAC) by the control with 0.60 uMTE/g. The GC/MS analysis showed 28 volatile constituents, withα-Terpinyl acetate, geranyl oleate, and oleic acid being the most abundant. DIC (140 °C and 30 s) and UAE showed the best yield and chemical profile. The SEM microscopy of untreated seeds revealed collapsed structures before the oil cell layer, which reduced the extraction yield, contrary to DIC-treated seeds, with more porous structures. Therefore, combining innovative extraction methods could solve the drawbacks of traditional extraction methods.

## 1. Introduction

Cardamom is a large perennial, herbaceous, rhizomatous monocot belonging to the *Zingiberaceae, Elettaria*, and the cardamom species (Maton) [1]. It is well known for its flavoring constituents and bioactive compounds of dried ripe fruits (capsules or pods), which are of trade importance to the cosmetic, food, and pharmaceutical industries [2]. In food industries, cardamom is widely used for flavoring beverages, processed foods, and confectionery. In cosmetic industries, it is used for fragrances, aromatherapy, and hair and skin care products, among other applications [3]. Moreover, cardamom has been used as an antimicrobial, antibacterial, and antioxidant in the pharmaceutical field, among other activities [4].

Cardamom essential oil possesses several biological activities such as antioxidant [5,6], anticarcinogenic, and antimicrobial, among others [6,7]. These activities are related to oxygenated monoterpenes and phenyl prostanoids [7]. The biological activities of cardamom chemical compounds have permitted its use in folk and modern medicine for controlling asthma, nausea, diarrhea, cataracts, teeth, gum infections, and digestive, kidney, and cardiac disorders [8].

*Elettaria cardamomum* fruits have been used as a flavoring ingredient and medicinal spice, being their essential oil one of the most common cardamom products [4]. Essential oils (EO) are a combination of various volatile lipophilic compounds (in some cases, more than 100) which give the plant its distinct aroma, taste, and flavor characteristics [9].

Specifically, in cardamom capsules and seed essential oils, it has been identified that the most flavor identity comes from the combination of 1,8-cineole, limonene, and α-terpinyl acetate. Moreover, in conjunction with these three flavors, α-pinene, β-pinene, terpineol, citronellal, linalool, and allo-aromadendrene represent cardamom’s essential flavor components of oil [10,11,12]. Moreover, as indicated by Mani et al. [13] these components are released from the cardamom during distillation based on its volatility; then, beyond the raw material, this composition can vary according to the selected essential oil extraction method.

The most common methods to obtain cardamom essential oil (CEO) are steam and hydrodistillation. However, some drawbacks of these techniques are low yields, long extraction kinetics, and the formation of undesirable compounds. As a result, it can give off-flavors and off-aromas (owing to the time of exposure to high temperatures) [14]. The Instant Controlled Pressure Drop, known by its French acronym DIC (Détente Instantanée Contrôlée), became a new technology to improve the essential oil extraction operation (yield and extraction time) and the overall quality of CEO [5,15].

The DIC is a thermo-mechanical process that consists of subjecting natural food matrices to saturated steam pressure treatments (100 to 900 kPa) for a few seconds, followed by an abrupt and controlled pressure drop towards a vacuum (10 to 5 kPa) at a rate higher than 500 kPa per second. The instant pressure drop creates autovaporization, texturing, and expelling processes. At its inception, the DIC technology was used to dry and texture products (known as swell drying). However, shortly after, its benefits on microstructural changes were also extended to EO’s extraction [15]. Thus, under suitable DIC treatment parameters (steam pressure, treatment time, initial moisture content of raw materials, among others), it is possible to increase both the porosity and tortuosity of biological matrices and, thereby, the EO’s extraction. This technique has allowed the intensification of the essential oil extraction of various products such as orange peels (*Citrus sinensis*), rosemary leaves (*Rosemarinus officinalis L.*), thyme (*Thymus capitatus*), myrtle leaves (*Myrtus communis L*.), lavender Grosso (*Lavandula intermedia var. Grosso*), and black cumin seeds (*Bunium persicum*) [15,16]. 

On the other hand, ultrasound-assisted extraction (UAE) has also been recognized for its potential in the phytopharmaceutical extraction industry for a wide range of herbal extracts [17]. Ultrasound generates cavitation, which can be explained by the production and breakdown of microscopic bubbles, which lead to cell membrane damage, providing a fast rate of extracted materials and a high yield [18]. Hence, coupling ultrasound to other techniques (such as hydrodistillation and microwave-assisted hydrodistillation) has enabled essential oil extraction from various biological matrices. Some examples are *Cinnamomum cassia* bark [19], *Perilla frutescens* (L.) Britt. leaves [20] and *Cymbopogon winterianus* leaves [21].

Then, if DIC pre-treatment and ultrasound-assisted extraction can independently increase the yield of essential oils, coupling these two methods could provide even better results. In this context, this study proposes to evaluate the effect of coupling the pre-treatment by DIC and ultrasound-assisted extraction on the cardamom essential oil (CEO) yield, the antioxidant activity by DPPH scavenging capacity and Trolox equivalent antioxidant capacity (TEAC), and chemical compositions by gas chromatography-mass spectrometry (GC-MS). Furthermore, to better understand the effect of the DIC pre-treatment on the microstructure of *Elettaria cardamomum* seeds, scanning electron microscopy (SEM) was used.

## 2. Results and Discussion

### 2.1. Essential Oil Yield

The effect of DIC treatment coupled with ultrasound-assisted extraction (DIC + UAE) on cardamom essential oil yield is shown in Table 1. Results showed that in almost all DIC and UAE treatments, the EO yield was increased from 1 to 44% in contrast to the control. The only exception occurred in DIC 6 (158 °C and 19 s), where the EO yield was lower than the control (14.278% vs. 15.596%). The highest extraction percentages were found on DIC 7 (140 °C and 30 s), DIC 3 (140 °C and 45 s), and DIC 5 (158 °C and 41 s), with 21.694%, 21.882%, and 22.526%, respectively. As can be observed in the Pareto chart (Figure 1a), the linear treatment time (t) and the interaction between the steam temperature and the thermal treatment time (T x t) crossed the reference line. It means that both factors impacted significantly on the EO yield (alpha level of 0.05). Moreover, regarding the surface response graph (Figure 1b), it can be remarked that the EO yield could be increased by applying a DIC treatment under high steam temperatures and high treatment times. The mathematical relationship obtained was a polynomial equation representing the quantitative effect of process variables and their interactions on the measured response.

On the other hand, by comparing the EO yield % obtained in this study to the results of Teresa-Martínez et al. [5] it can be remarked that by coupling DIC and UAE, the EO yields were dramatically increased concerning DIC and hydrodistillation. While the highest yield obtained by DIC and hydrodistillation was 4.4% (DIC: 140 °C and 30 s), the highest yield of DIC and UAE was 22.5% (DIC: 158 °C and 41 s). This significant increase in the EO yield could be attributed to the supplementary effects of both technologies: first, the DIC as a texturing pre-treatment opened the cells, enabling an easy diffusion of EO, and secondly, the ultrasound generated cell oil disruption and improved solvent penetration and the capillary effect [15,22]. 

It is worth noting that cardamom EO yields obtained in this study overtake the generated values of other studies under hydrodistillation and UAE, ranging between 1% and 9.5% [4,22,23]. Therefore, it could be concluded that employing DIC as a pre-treatment before the extraction methods allowed the swelling of cells. As a result, it generates higher extraction yields for the CEO. 

Finally, Allaf et al. [15] also indicated that DIC technology could directly extract essential oils by applying numerous DIC cycles on the raw material matrix using steam as a solvent. Then, in future studies, it could be interesting to evaluate the cardamom EO yields and their quality by only using this emerging technology.

### 2.2. Antioxidant Activity of CEO

#### 2.2.1. DPPH Free Radical Scavenging Capacity 

The results of the DPPH assessment of the CEO expressed as % discoloration are shown in Table 1. The most potent antioxidant activity was obtained in the treatment DIC 2 (165 °C, 30 s, 0.70 MPa) with a 79.48% discoloration vs. 57.75% of control. Then, by comparing the DPPH scavenging activity of essential oil extracted from DIC and UAE vs. only UAE, an increase of nearly 22% can be observed. In contrast, the lower DPPH scavenging activity was obtained by the DIC 10 (140 °C, 30 s, 0.36 MPa) with 44.30% discoloration. Regarding the Pareto chart (Figure 2), neither the temperature nor thermal processing time could explain the antioxidant activity changes under the selected DIC studied parameters.

According to previous studies, the maximum DPPH scavenging activity obtained from the CEO through steam and hydrodistillation was around 70% [24,25]. On the other hand, Teresa-Martínez et al. [5] showed that the CEO obtained by coupling DIC as a pre-treatment before hydrodistillation (HD) increased its DPPH scavenging capacity. DIC and HD (165 °C and 30 s) performed 86% DPPH inhibition, while the control (only HD) performed 57.02%. The obtained results corroborate both facts, the antioxidant potential of the CEO and the positive impact of DIC on the DPPH scavenging activity. Similarly, Namir et al. [26] reported that after DIC treatment, the DPPH radical scavenging activity of cactus pear peel increased to 53%. Also, Mkaouar S. et al. [27] found that DIC-treated olive leaf extracts presented the highest DPPH value (95.7%). In contrast, commercial butylated hydroxytoluene (BHT) showed a DPPH scavenging capacity of 88.4%. The DIC is a thermomechanical process characterized by a short time of exposure to high temperatures and water autovaporization, which guarantees rapid cooling and swelling, preventing the thermal degradation of secondary metabolites. 

#### 2.2.2. ABTS Trolox Equivalent Antioxidant Capacity Determination (TEAC)

Contrary to DPPH scavenging activity, in which a maximum value is desirable, in this essay, the best antioxidant capacity is represented by lesser TEAC values. Then, as can be observed in Table 1, the control presented the best antioxidant capacity with 0.60 uMTE/g, followed by DIC 6 (158 °C, 19 s, 0.58 MPa) and DIC 1 (140 °C, 30 s, 0.36 MPa), with 0.75 uMTE/g and 0.81 uMTE/g, respectively. In Figure 3, the Pareto chart showed that under the selected DIC studied parameters the TEAC was not influenced by the temperature or the thermal processing time.

According to Al-Zereini, W. et al. [7] and Wang, H. F. et al. [28] the TEAC values in non-treated cardamom seeds are negligible. However, by comparing the TEAC values of this study to the results of Teresa-Martínez et al. [5] who evaluated the TEAC of CEO extracted by DIC and HD and only HD (control), it can be remarked that there was a significant enhancement of the antioxidant activity of CEO by UAE. 

While the TEAC value of essential oils extracted by only HD was 13.66 uMTE/g EO, only UAE triggered 0.60 uMTE/g. It could be suggested that the cavitation bubbles generated during the propagation of the ultrasonic waves triggered disruption in the cardamom cells, which allowed a better extraction of the hydrophilic and lipophilic antioxidant compounds. It is beneficial in protecting the essential oil components, sensitive to heat, and offers new profiles of EO where the abundance of compounds varies according to the experimental conditions. 

Moreover, to better understand the effect of DIC treatment on the TEAC of essential oils, further research studies are needed to explore the effect of other variables, such as the initial moisture content and the number of DIC cycles. 

Moreover, even though neither temperature nor processing time affects TEAC or DPPH scavenging capacity (according to the Pareto charts), this does not mean a drawback since we can take advantage of the tendencies in both factors without affecting the antioxidant properties of the essential oil.

### 2.3. Gas Chromatography Analysis (GC-MS) of CEO

GC-MS was used to evaluate the influence of only UAE (control) and DIC and UAE on the chemical profile of CEOs as an exploratory approach aiming to identify the leading families of compounds presented in each treatment. The list of constituents identified among the different treatments is presented in Table 2, with the corresponding abbreviation employed for the statistical analysis and the heatmap (Figure 4). In total, 28 volatile constituents were identified among all the samples. Monoterpenoids, sesquiterpenoids, and polyunsaturated and saturated fatty acids were the most abundant compounds. These results were similar to Jena et al. [29] who identified oxygenated monoterpenes as the most abundant class of compounds (23.53%). It is followed by sesquiterpene hydrocarbons (22.64%), monoterpene esters (20.71%), oxygenated sesquiterpenes (7.33%), sesquiterpene oxides (7.13%), monoterpene hydrocarbons (6.21%), and monoterpene aldehyde (4.57%). 

It is worth mentioning that some monoterpenes show antioxidant activity, as described by Tabaszewska et al. (2021) and Al-Zereini1 (2022), which could be related to this property observed in the present study [7,30]. 

The GC-MS results are presented as a heatmap to determine the differences between the oil composition among the treatments (control and 13 DIC and UAE samples). Figure 4 shows a heatmap profiling the compounds of CEO and their abundance, where compounds results are shown as the percentage of area % normalized. In the heatmap, the range of colors goes from green to red; the higher the percentage of area, the greener the color. On the contrary, the lower the area percentage, the redder the box. Furthermore, in the case of the absence of the compound, the red boxes mean no presence.

By analyzing the heat map (in Figure 4), it can be emphasized that the most abundant compounds were TAC (α-Terpinyl acetate), GO (geranyl oleate), OLAD (oleic acid), DOH (1,6,10-Dodecatrien-3-ol, 3,7,11-trimethyl-, (E)-), and SOH (ç-Sitosterol). Similar results were shown by Al-Zereini et al. [7] who found that α-Terpinyl acetate was the major component of CEO (55.99%), followed by 1,8 cineole (8.82%), linalool (6.99%), dihydrocarveol (6.06%), geraniol (4.46), Z-caryophyllene (3.82%), E-nerolidol (3.07%), eugenol (2.31%), and terpinen-4-ol (1.83%).

Among the monoterpenes, TAC was present in all the samples, showing that DIC and UAE preserves this compound. In fact, under DIC 1, DIC 2, DIC 3, DIC 5, DIC 8, DIC 9, DIC 10, DIC 11, and DIC 13 treatments, the normalized area of TAC was 100%. However, in DIC 4, DIC 6, DIC 7, and DIC 12, the % of TAC was reduced compared with the control, DIC 4 (140 °C and 30 s), and DIC 12 (140 °C and 15 s), the treatments that performed the lowest percentages of TAC. In Appendix A, it can be remarked that these changes in the TAC profile could not be explained by the selected DIC studied variables (steam temperature and thermal treatment time). As indicated by Mahanta, BP et al. [31], monoterpenoids, such as α-Terpinyl acetate, belong to the significant thermolabile aroma chemicals. Therefore, the TAC degradation possibly was linked to a thermal rearrangement.

Similarly, GO was present in all the samples. In Figure 4, it is possible to appreciate that the CEO obtained by DIC and UAE has higher percentages of this compound than the control. Specifically, DIC 4 (140 °C and 30 s), DIC 8 (122 °C and 19 s), and DIC 12 (140 °C and 15 s) were at least double the percentage compared to the control. In Appendix A, it can be remarked that changes in the GO profile could not be explained by the selected DIC studied variables (T and t). Then, it can be concluded that this polyunsaturated fatty acid was well preserved under the selected ranges of T and t.

On the other hand, OLAD was present in almost all treatments except for DIC 2 (165 °C and 30 s). Specifically, by comparing the control (5.48%) to DIC and UAE samples (45.27 to 100%), a drastic increase can be remarked. For this compound, DIC 10 (140 °C and 30 s) and DIC 11 (115 °C and 30 s) showed lower concentrations (46.59% and 45.27%, respectively), and DIC 4 (140 °C and 30 s), DIC 6 (158 °C and 19 s), DIC 7 (140 °C and 30 s), and DIC 12 (140 °C and 15 s) showed the highest concentrations (100% in all the cases). It is worth noting that under the DIC central point treatment (140 °C and 30 s), the oleic acid concentration varied. Regarding the results of the five central points (DIC 1, DIC 4, DIC 7, DIC 10, and DIC 13), it can be remarked that the average of all treatments was 78.45%, being 14 times bigger than the control concentration.

Meanwhile, DOH was absent in the control and DIC 4 (140 °C and 30 s), DIC 5 (158 °C and 41 s), and DIC 13 (140 °C and 30 s) treatments. However, it was abundant in DIC 7 (140 °C and 30 s) at 94.87%, followed by DIC 6 (158 °C and 19 s) at 77.26%, DIC 8 (122 °C and 19 s) at 75.96%, and DIC 9 (122 °C and 41 s) with 72.03%.

The phytosterol SOH was present in almost all the treatments except for DIC 1 (140 °C and 30 s) and DIC 2 (165 °C and 30 s). The control performed a SOH concentration of 7.0%, and DIC 7 (140 °C and 30 s), DIC 4 (140 °C and 30 s), and DIC 12 (140 °C and 15 s) showed the highest concentrations (18.28%, 17.00%, and 17.00%, respectively).

Upon further analysis, SIT was present in only four extracts (DIC 6, DIC 7, DIC 10, and DIC 11) with concentrations between 3.48 and 7.71%. This ketone was absent in control. Regarding PA, this monoterpenoid was only present in the treatment DIC 12 (140 °C and 15 s) with a concentration of 6.12%. Likewise, Z-DDP, P-HNP, OA, OCAD-E, DAT, 2-HC, DOHT, 3-TT, DOH, LAC, and GEOH were present in at least one DIC treatment and utterly absent in the untreated seeds.

Furthermore, concerning TEOH concentration, one of the most characteristic compounds in cardamom seeds, it can be highlighted that untreated samples showed a concentration of 20.96%. In the case of DIC and UAE samples, the concentrations ranged between 6.39% and 26.20%, and were absent in DIC 4, DIC 7, and DIC 12 samples. The lowest concentration was shown by DIC 6 (158 °C and 19 s) extracts, and the higher concentrations by DIC 10 (140 °C and 30 s) and DIC 11 (115 °C and 30 s) with 25.94% and 26.20%, respectively. Hence, by comparing the obtained results to the study of Noumi et al. [2] who evaluated the components of cardamom essential oils obtained by hydrodistillation by GC-MS, DIC and UAE allowed an increase to 8.4 times the concentration of α-Terpineol (3.1% vs. 26.20%) [2]. Moreover, UAE alone allowed an increase of 6.7 times concerning hydrodistillation.

Contrastingly to TEOH, DIC and UAE treatment negatively impacted OCAD. For this polyunsaturated fatty acid, the control showed a concentration of 76.88% vs. DIC treatments with concentrations from 12.08% to 29.88%. The DIC 8 (122 °C and 19 s) showed the highest concentration. Appendix A shows that under the selected DIC studied variables, neither the steam temperature nor the treatment time impacted significatively on the OCAD concentration. However, regarding the results, it can be concluded that OCAD has low thermal stability.

Another compound present in all the samples was n-Hexadecanoic acid (HEXAD). For this fatty acid, the control showed a concentration of 66.96%. Moreover, the DIC and UAE samples presented values between 33.24% and 76.81%; DIC 7 (140 °C and 30 s) was the treatment with the highest concentration. Regarding the Pareto chart in Appendix A, it can be remarked that changes in the HEXAD profile could not be explained by the selected DIC studied variables (T and t).

Regarding SAC, DIC and UAE showed concentrations between 8.21% and 18.20%. However, by comparing these results to the control (25.84%), it can be remarked that under DIC-selected conditions, the concentration of SAC was reduced. Observing the Pareto chart in Figure 5, the interaction between steam pressure (T) and thermal treatment time (t) impacted SAC concentration. Additionally, regarding the response surface graph, it can be concluded that the SAC concentration could be increased under two conditions: (1) low temperatures (~110 °C) and treatment times of around 50 s or (2) high temperatures (~170 °C) and treatment times of around 10 s.

Therefore, 8-Acetoxycarvotanacetone (ACA) was the unique compound presented only in the control with a concentration of 28.96%. It can be concluded that this compound was degraded under the selected DIC and UAE treatment conditions. In the future, it will be interesting to evaluate the thermal behavior of this menthane monoterpenoid under lower DIC temperatures.

Finally, regarding the individual compounds profile of each sample, it can be remarked that under the selected DIC studied variables (T and t), it was possible to trigger different chemical profiles. Then, according to the final application of the CEO, it is possible to determine the optimal DIC operative conditions to increase a target compound. This way, if the target compounds are TEOH and TAC, the optimal treatment will be DIC 11 (115 °C and 30 s) and UAE. However, suppose one is seeking a CEO with high concentrations of volatile compounds. In that case, DIC 7 (140 °C and 30 s) and UAE could be the option by showing high concentrations for DOH, HEXAD, P-HNP, GO, SOH, and SIT. Although TEOH was absent and TAC was lower in this treatment, it allowed for preserving or even increasing the concentration of various volatile compounds concerning the control. Furthermore, to better understand the relationship between the experimental conditions (DIC and UAE and only UAE) and the cardamom essential oil composition, a principal component analysis (PCA) was used.

### 2.4. Principal Component Analysis (PCA) of CEO Composition

The principal component analysis (PCA) of the cardamom essential oil composition is shown in Figure 6. In this figure, it can be observed that the formation of three components together explains 78.5% of the data’s variability. The data are presented by three principal components defined as a linear combination and correlation between each other. This PCA analysis improved discrimination and group classification in the three-dimensional visualization (Figure 6) [16]. Likewise, a well-defined classification of groups was distinguished in the two-dimensional visualization. Therefore, it reveals clusters of the observed variables in terms of their similarities. Hence, group compounds can be observed to correlate with each other; the correlation was detected between compounds labeled SAC, OCAD, and HMAC2.

In the same way, compounds with the labels TAC, TEOH, DOH, DAT, and GEOH correlate with each other, and DOT, HMMC, OLAD, HC, GO, and SOH form another group in the loading plot graph. It can be observed that the compound labeled HEXAD correlates more with the latter. In addition, a negative correlation between HEXAD and the compounds TAC, TEOH, HMAC2, SAC, and OCAD can be highlighted, i.e., while the latter concentration was low, HEXAD concentration increased.

Moreover, Figure 7 shows the score plot or distribution of the treatments in the same plane of components 1 and 2, where three groups are formed. In the first quadrant, group A is formed only by the control, group B comprises the treatments DIC4, DIC12, and DIC7, and finally, group C is composed of the rest of the treatments.

Lastly, the biplot graph found in the sAppendix A shows the detected compounds’ individual factors (treatments). Thus, it is observed that group A positively correlates with the highest amounts of compounds such as SAC (Sobrerol 8-acetate) and OCAD (9, 12-Octadecadienoic acid), where most of the treatments are included. Secondly, group B of the treatments correlates with the highest parts of HEXAD (n-Hexadecanoic acid), SOH (ç-Sitosterol), and OLAD (oleic acid). However, the latter two are more related to the DIC 7 treatment. Furthermore, the compound HEXAD correlates with groups B and A of the treatments. Therefore, it can be established that treatments DIC 4, DIC 12, DIC 7, and the control were more efficient for extracting fatty acids or those with a non-polar tendency.

The results in studies employing traditional extraction methods (such as hydrodistillation, steam distillation, and solvent extraction) have demonstrated abundance in a wide variety of terpenoids (specifically SAC, OCAD, and HMAC2). For example, using only hydrodistillation, they obtained 1,8-cineole as the main compound [2,32]. On the other hand, Chen et al.’s [33] research employs direct steam distillation extraction and reported eucalyptol and β-pinene as the most abundant compounds in CEO [33]. Furthermore, current investigations employing novel techniques (such as solid phase microextraction, thermal desorption, microwave-assisted extraction, supercritical fluid extraction, and ultrasound-assisted extraction) have shown similar results to this study. For example, they have found α-Terpinyl acetate as the primary component of cardamom EO, followed by 1,8 cineole, linalool, dihydrocarveol, geraniol, Z-caryophyllene, E-nerolidol, eugenol, and terpinen-4-ol [7,29,34,35].

To sum up, it is essential to remark that the principal component analysis of cardamom essential oil composition could be considered an important tool to select better the DIC and UAE operative extraction conditions to increase the concentration of target compounds.

### 2.5. Scanning Electron Microscopy (SEM)

Cardamom capsules contain around 15 to 20 aromatic seeds embedded in mucilage and enclosed in a semi-hard fleshy green to yellow pericarp. After drying, the pericarp becomes fibrous, and the mucilage disappears [36]. Dried cardamom seeds have a hard seed coat composed mainly of three polysaccharides (cellulose, hemicellulose, and pectin) which play a protective cover role for seeds [37]. Microscopic studies reveal that the main parts of cardamom seeds from the outside to the inside are (a) a thin transparent arillus, (b) an epidermis, (c) an oil cell layer, (d) a sclerenchymatous layer, (e) a perisperm, (f) an endosperm, and (g) an embryo [38,39]. The oil cell layer is composed of large rectangular cells filled with globules of volatile oil [36].

In this study, the scanning electron micrographs allowed us to evaluate the effect of the different DIC treatments on the microstructure of cardamom seeds. Figure 8 and Appendix A show the SEM micrographs of whole and cross-section cuts of DIC-treated and untreated (control) cardamom seeds.

The SEM microscopic images (whole and cross-section cut) of untreated cardamom seeds reveal that several collapsed cells exist before the oil cell layer (arillus, epidermis, and outer parenchyma). This collapsed structure reduces the essential oil’s effective diffusivity and extraction yield. Then, even if it is well known that raw material grinding increases EO yields due to an increase in the specific surface exchange, this operation could be improved through texturing processes such as DIC treatment [15]. Therefore, SEM microscopic images of DIC-treated seeds showed that under the selected ranges of saturated steam temperature and thermal processing time, it was possible to modify the microstructure of cardamom seeds. In this respect, it seems that the microstructure expansion of cardamom seeds was mainly linked to the steam condensation that occurred during the saturated steam injection into the reactor and to the autovaporization of seed water. Furthermore, on SEM micrographs of the whole cardamom-treated seeds, it can be observed how DIC treatment generated fissures in the outer layer of seeds (especially in DIC 1, DIC 11, and DIC 12 samples).

On the other hand, regarding cross-section cuts (Appendix A), more porous structures can be observed on DIC-treated seeds, specifically in DIC 1, DIC 3, DIC 5, DIC 6, DIC 9, and DIC 11. Moreover, to better understand the impact of DIC studied parameters (steam temperature and treatment time) on the absolute expansion ratio of cardamom seeds, it can be suggested to determine the intrinsic and apparent density of dried seeds in future studies. Moreover, X-ray microtomography and 3D image analysis could be applied to accurately calculate the volume and the area of seeds [40].

## 3. Materials and Methods

### 3.1. Plant Material and Chemicals

The cardamom seeds (*Elettaria cardamomum*) were obtained from Argovia Farm, located in the Soconusco, Chiapas. The main geographical characteristics of Argovia farm are as follows: minimum height of 500 m, maximum height of 800 m, 15°02′48” latitude, 15°02′48” longitude, average temperature of 23 °C, and an average rainfall of 5500 mm/year. 

Chemical reagents for antioxidant activity, 2,2-diphenyl-1-picrylhydrazyl (DPPH), (±)-6-Hydroxy-2,5,7,8-tetramethylchromane-2-carboxylic acid (Trolox), and 2,2′-azinobis(3-ethylbenzothiazoline-6-sulfonic acid) (ABTS) were obtained from Sigma–Aldrich^®^ (St. Louis, MO, USA). The industrial ethanol (solution of 96%) and the double-distilled water employed in the sonication and the GC-MS were retrieved from CTR Scientific^®^ (Monterrey, Mexico). The helium employed for the GC-MS analysis was ultra-high-purity Helium 5.0 gas obtained from Linde^®^ (Monterrey, Mexico).

### 3.2. Essential Oil Extraction

Aiming to obtain cardamom essential oils (CEO), first, cardamom seeds were divided into 14 lots (100 g each): 13 were used to evaluate the effect of coupling the DIC treatment to UAE, and the last 1 was used to evaluate the effect of UAE on raw material without DIC pre-treatment. First, the 13 samples were pretreated under the conditions described in Section 3.2.1, and later, essential oils were extracted as described in Section 3.2.2.

#### 3.2.1. Instant Controlled Pressure Drop (DIC) Treatment

The DIC treatment of cardamom seeds was carried out in a DIC lab-scale equipment (ABCAR-DIC Process—La Rochelle, France), following a DIC cycle of five steps. For the first step, seeds were introduced into the reactor, followed by pressure drops towards a vacuum (step 2), followed by subjection to high-pressure saturated steam (0.17–0.7 Mpa) for some seconds (15–45 s) (steps 3 and 4). Finally, there was an instant controlled pressure drop close to a vacuum (step 5). After DIC treatment, cardamom seeds were stored at ambient temperature (22 °C) until further analysis.

Preliminary experiments identified the most relevant ranges for DIC operating parameters (T = steam processing temperature and t = thermal processing time). For this experiment, a central composite rotatable experimental design was defined with two factors (T and t) and five levels (−α; −1; 0; +1; +α), which led to 12 experiments (4 factorial points, 4-star points, and 4 central points) (Table 3).

#### 3.2.2. Ultrasound-Assisted Extraction for Cardamom Essential Oil

First, the DIC-pretreated cardamom seeds were ground in a porcelain mortar until a fine powder was obtained. After that, the extraction solution was prepared according to the modified protocol described by Chen et al. 2021 [21] and Morsy et. al. 2015 [23]. Ethanol:water (90:10 *v*/*v*) solution was used, and subsequently, 5 g of cardamom seed powder was mixed with 30 mL of the extraction solution in a 50 mL Corning^®^ (New York, NY, USA) tube. Then, the sonication stage was carried out with a standard titanium horn (½ inch diameter) in cycles of 19 min in pulse/pause mode. The pulse duration was 5 min to avoid overheating. The resting interval was 2 min with a 50% amplitude. After the UAE, the obtained samples were filtered and concentrated. First, each solution was filtered using a Whatman^®^ qualitative filter paper, Grade 6 (3 μm, circles, diam. 90 mm), and then put in 50 mL amber-glass tubes. After that, samples were concentrated under evaporation cycles of 6 h with a maximum temperature of 45 °C. The final extracts were poured into 10 mL amber-glass tubes and refrigerated at 4°C until further use. The types of equipment used for sonication and concentration of extracts were a Digital Sonifier ^®^ Model S-450D, Branson, MO, USA, and a vacuum evaporator 4D Rocket Synergy™ from GENEVAC^®^, Pennsylvania, USA, respectively. The yield was calculated as grams of recovery oil per 5 g of seed powder and expressed as a percentage (%).

### 3.3. Scanning Electron Microscopy (SEM) of Cardamom Seeds

Dry seeds of raw material and DIC-treated samples were mounted on SEM specimen stubs with short pins (12.5 mm dia, 3.2 × 8 mm) using a double-sided adhesive carbon type. Subsequently, the seeds were coated with 5 nm pure gold using a Q Series Rotary Pumped Coating System (Quorum^®^ Q150R ES). For each treatment, at least 84 seeds were randomly selected and studied, and only the most representative samples were chosen. The coated materials were examined and photographed with a Zeuss^®^ Evo MA25 scanning electron microscope at an accelerating voltage of 8.0 kV and with a high vacuum. For the detection mechanism related to imaging, the Signal SE1 detector was implemented. Two micromorphological analyses were performed to study the effect of each treatment on the cardamom seeds. First, the whole-seed structure was analyzed. Secondly, the internal morphology of the seeds was examined through a cross-section cut (made by placing a sterilized needle in the center of each seed to make a clean cut).

### 3.4. Compound Profile Analysis of Cardamom Essential Oils by Gas Chromatography Coupled to a Mass Spectrometer Detector (GC-MS)

First, a dilution was made for each sample, 1 mL of CEO and 9 mL of HPLC-grade hexane (1:10 *v*/*v*). Then, aliquots of 1 µL for each sample were injected automatically on a Perkin Elmer gas chromatograph Clarus coupled with a mass spectrometer (MS), model Clarus 690, and Clarus SQ8T. The GC method was performed on a TG-5 MS (30 m × 0.25 mm × 0.25 μm) capillary column with helium as the carrier gas at a flow rate of 1 mL per minute. The oven temperatures were initiated at 70 °C for 2 min, then increased at 10 °C per minute to 200 °C and held for 5 min. The total run time was 35 min and 33 s with an injector temperature of 256 °C. The MS analysis was carried out with the following parameters: electron ionization ion source, electron energy 70 eV, the temperature of quadrupoles 150 °C, the temperature of interface 200 °C, and m/z = 30–550 amu. Finally, the compounds were identified using the mass spectra library NIST Copyright© 1998 (the chromatograms obtained are shown in Appendix A). As an exploratory approach, no standard was required, and hexane (HPLC grade) was used as a blank. The data retrieved were used for qualitative analysis. Specifically, the signals of the identified compounds were compared with the MS library. The aim was to obtain a preliminary profile of the CEO of each treatment.

### 3.5. Antioxidant Capacity

#### 3.5.1. DPPH Free Radical Scavenging Capacity

First, 0.02 mL of essential oil solution (460 mg/mL) was added to a 96-well flat-bottom plate containing 0.2 mL of DPPH solution (125 µM DPPH in 80% methanol) to determine the antiradical activity of DPPH. Afterward, the plate was covered and left for 90 min in the dark at room temperature. Subsequently, the plate was read at 520 nm in a visible-UV microplate reader (680 XR Microplate Reader, Bio-Rad Laboratories, Inc). The DPPH scavenging activity was calculated as a percentage of DPPH discoloration using Equation (1):(1)$$\mathrm{DPPH}\;\%\;\mathrm{discoloration}=100\;\times \left(\frac{1-A\;\mathrm{sample}}{A\;\mathrm{control}}\right)$$
where *“A sample”* is the absorbance of the sample and *“A control”* is the absorbance of the control (DPPH in the absence of antioxidants) [41]. All the samples were prepared in triplicate.

#### 3.5.2. Trolox Equivalent Antioxidant Capacity (TEAC)

The Trolox equivalent antioxidant capacity (TEAC) method is based on the ability of an antioxidant to scavenge the preformed radical cation ABTS•+ relative to that of the standard Trolox. The total antioxidant capacity of extracts was determined according to Re et al.’s protocol [42]. Briefly, the ABTS•+ radical cation was produced by reacting 7 mM of 2,2′-azinobis(3-ethylbenzothiazoline- 6-sulfonic acid) diammonium salt and 2.45 mM potassium persulfate after incubation at room temperature in the dark for 16 h. The ABTS•+ solution was diluted with ethanol to an absorbance of 0.80 ± 0.1 at 734 nm. For the TEAC analysis, 0.2 mL of the ABTS reagent and 0.02 mL of sample extracts were mixed thoroughly with a microwell in a 96-microwell plate. The absorbance readings were taken at 734 nm after 6 min of incubation using a visible-UV microplate reader (680 XR Microplate Reader, Bio-Rad Laboratories, Inc.). Trolox standard solutions in methanol were prepared and assayed under the same conditions (0–700 µM). Moreover, the TEAC of the sample was calculated as follows: the µM of Trolox required to have the same discoloration degree as the samples.

### 3.6. Statistical Analysis

Statistical Software 2017 (TIBCO Software Inc., Palo Alto, California, CA, USA) was used for the statistical analysis. For the experimental design of the DIC treatment, statistical analysis was performed through the Pareto chart and the surface response methodology. First, the Pareto chart was used to identify the impact of variables on the responses. The vertical line in the Pareto chart determines the statistically significant effects at the 95% confidence level. Next, steam processing temperature (°C) and thermal processing time (s) were studied as independent variables.

Likewise, the impact of DIC treatment on essential oil yield extraction (%), antioxidant capacity (AOX in %), Trolox equivalent antioxidant capacity (TEAC in uMTE/g EO), and the content of some essential volatile compounds (% area) such as α-terpinyl acetate (TAC), sobrerol 8-acetate (SAC), n-hexadecanoic acid (HEXAD), 9,12-octadecadienoic acid (Z, Z)- (OCAD), and geranyl oleate (GO) were evaluated by using the response surface methodology (RSM).

On the other hand, to evaluate the impact of coupling the DIC treatment to UAE on the chemical composition of essential oils, the multivariate analysis of principal component analysis (PCA) was performed on GC-MS results. The multivariate analysis of Principal component analysis (PCA) with three axes was performed using SPSS (Version 19, IBM Corp, and Chicago, IL, USA). On the other hand, the two-dimensional PCA was performed with Excel 2019 (V17.0) from the package Microsoft 365®.

## 4. Conclusions

Traditional extraction methods (hydrodistillation, steam distillation, and organic solvent extraction) showed drawbacks such as losses and degradation of volatile compounds and low yields. Therefore, DIC and UAE were coupled and evaluated on the cardamom essential oil extraction. First, this study revealed that by coupling DIC and UAE, it was possible to increase the EO yield, being 22.53% vs. 15.6% (control). Also, UAE, as an extraction method, offered new profiles of EO where the abundance of compounds varied. The DIC treatment under the conditions of 140 °C and 30 s and UAE showed the best yield and chemical profile, according to the PCA. Regarding the SEM results of DIC-treated seeds, it can be concluded that DIC allowed the swelling of cells that generates higher CEO extraction yields. Additionally, the chemical profile of the CEO obtained by GC-MS allowed us to identify 28 volatile constituents, α-Terpinyl acetate, geranyl oleate, and oleic acid being the most abundant. Therefore, due to their relevance as a nutraceutical, this could be an optimal technique focused on Oleic acid extraction from cardamom seeds. However, quantitative research must evaluate this technique’s real potential before presenting it as an exemplary method. Finally, this study showed that combining innovative extraction methods (DIC and UAE) can solve the drawbacks of traditional extraction methods.

## Figures and Tables

**Figure 1 molecules-28-01093-f001:**
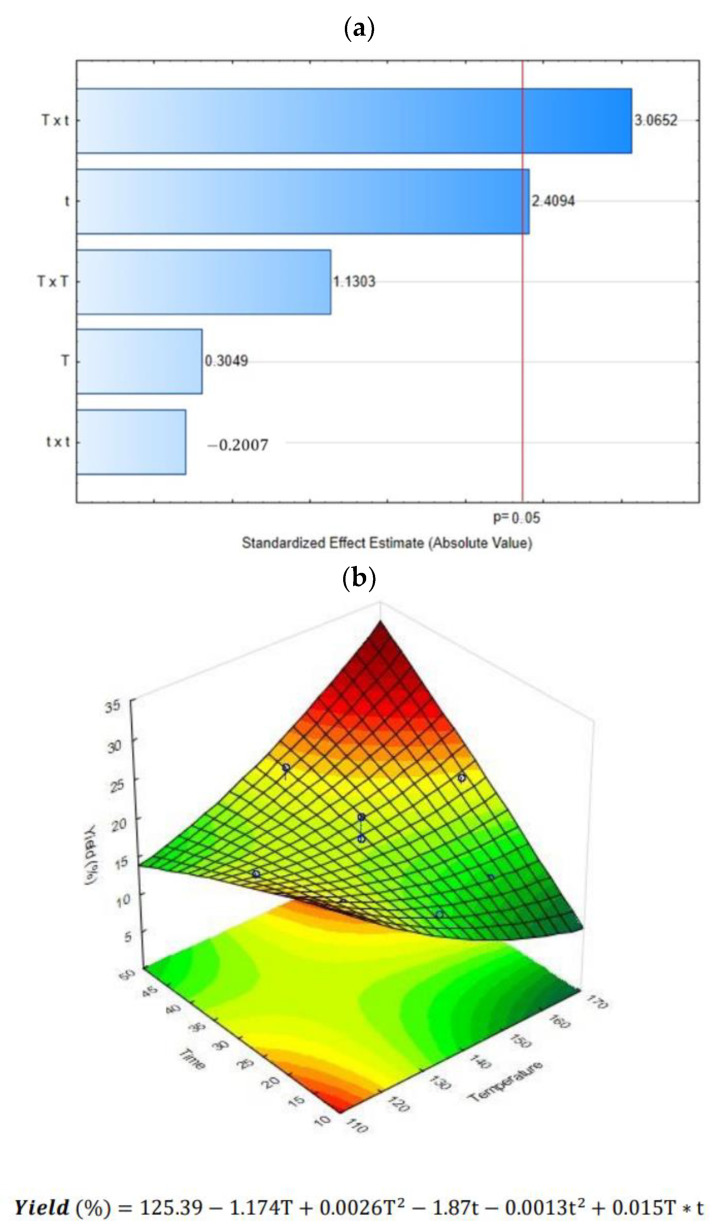
Effect of steam processing temperature “T” (°C) and thermal processing time “t” (s) on CEO yield (%): (**a**) Pareto chart and (**b**) Surface response analysis.

**Figure 2 molecules-28-01093-f002:**
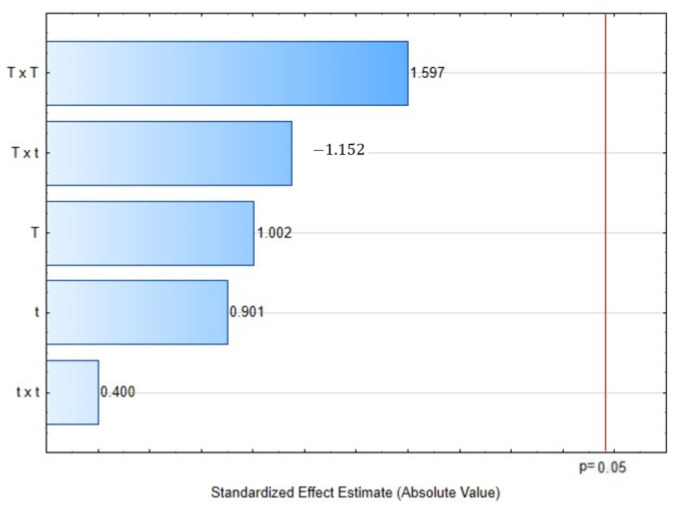
Effect of steam processing temperature “T” (°C) and thermal processing time: “t” (s) on DPPH scavenging capacity (% discoloration) of CEO: Pareto chart.

**Figure 3 molecules-28-01093-f003:**
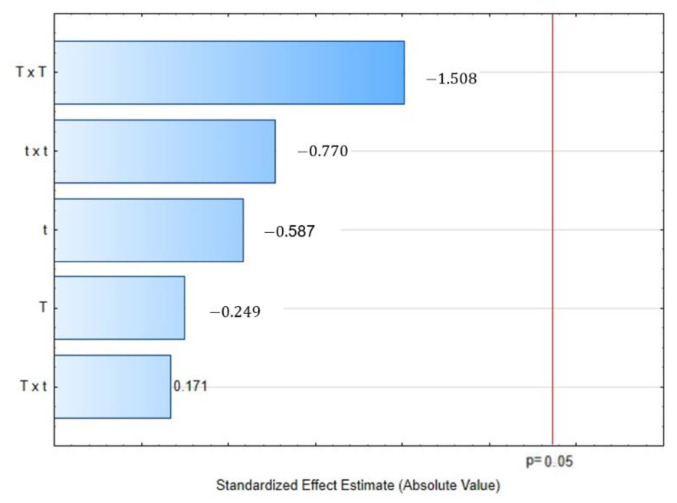
Effect of steam processing temperature “T” (°C) and thermal processing time “t” (s) on TEAC (uMTE/g EO) of CEO: Pareto chart.

**Figure 4 molecules-28-01093-f004:**
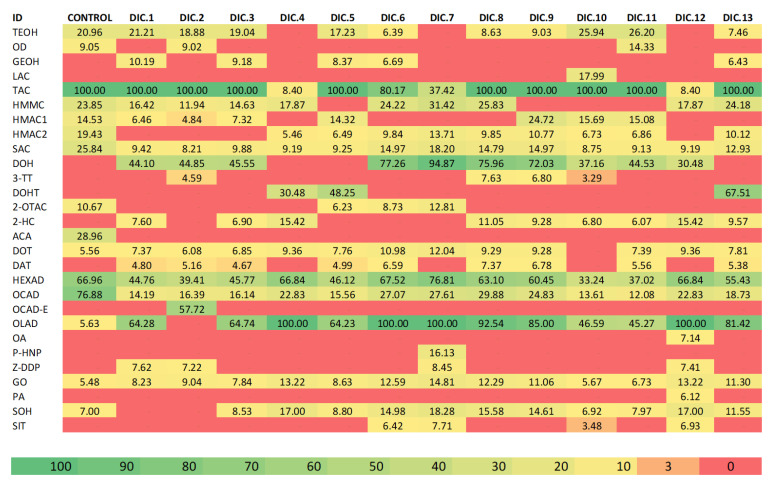
A heatmap profiling the compounds and their abundance for each treatment. The percentage of area % was normalized to obtain the profile of each compound. The color scale on the bottom of the figure explains the range and abundance.

**Figure 5 molecules-28-01093-f005:**
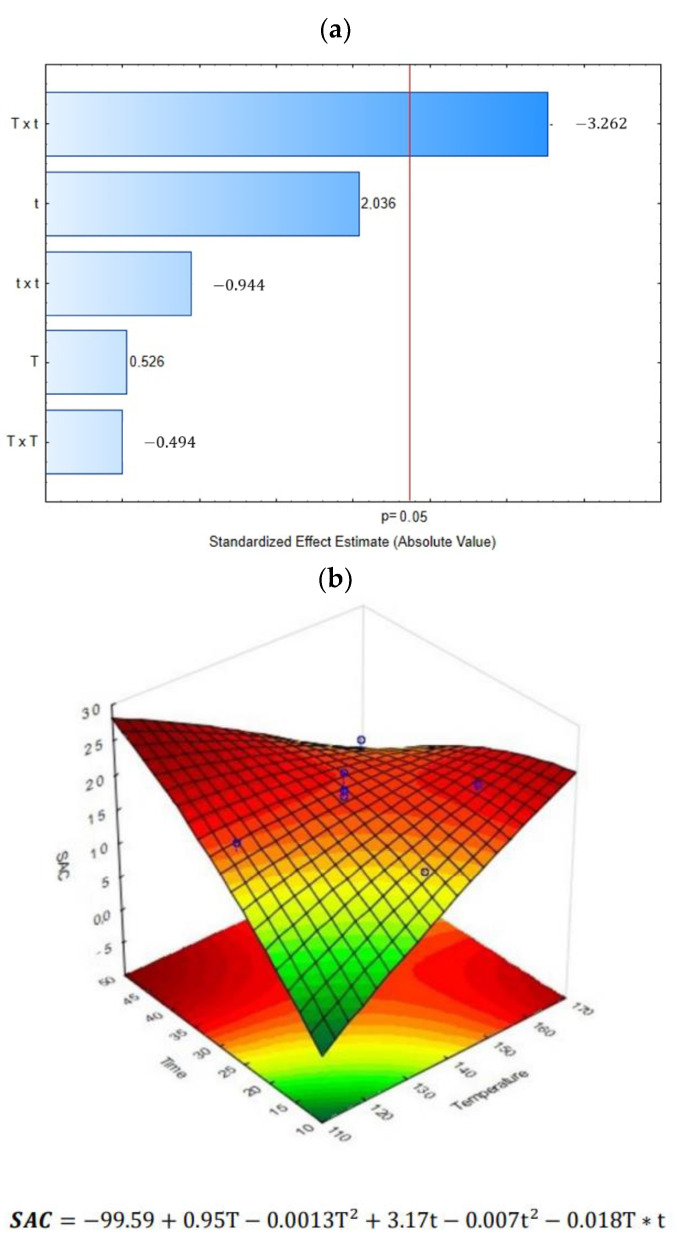
Effect of steam processing temperature “T” (°C) and thermal processing time “t” (s) on SAC (%) of EO: (**a**) Pareto chart and (**b**) Surface response analysis.

**Figure 6 molecules-28-01093-f006:**
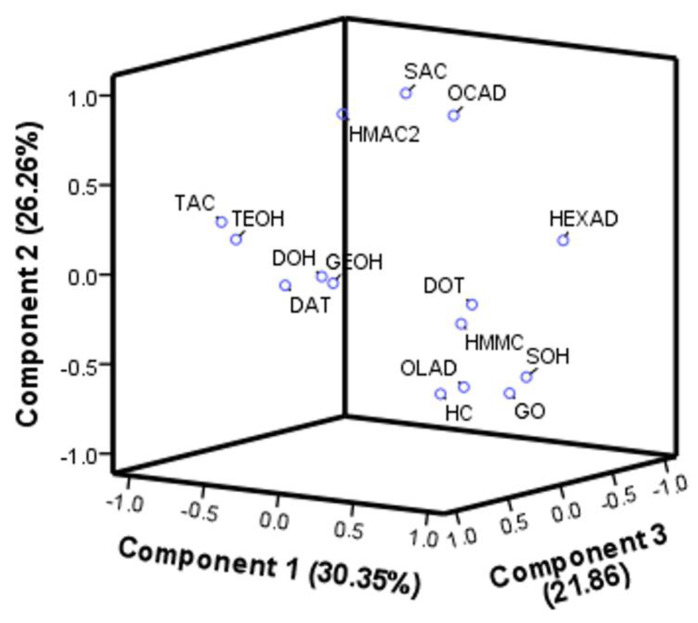
Loading plot of the main components. Three components were considered for the plot due to the distribution in the summary.

**Figure 7 molecules-28-01093-f007:**
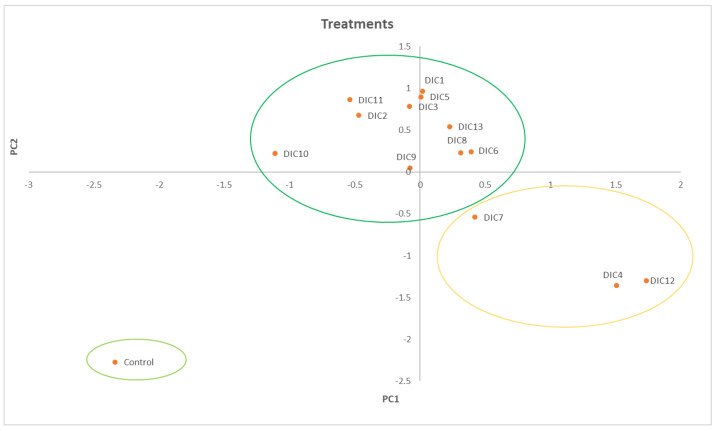
Analysis of Treatments. Projection of the distribution of the treatments (score plot) in the same plane of the first and second principal components. Group A, in light green, is formed only by the control. Group B (circled in yellow) is formed by DIC 4, DIC 12, and DIC 7. The last one is included because of its position in the fourth quadrant. Furthermore, Group C (darker green circle) is formed with the rest of the DIC treatments.

**Figure 8 molecules-28-01093-f008:**
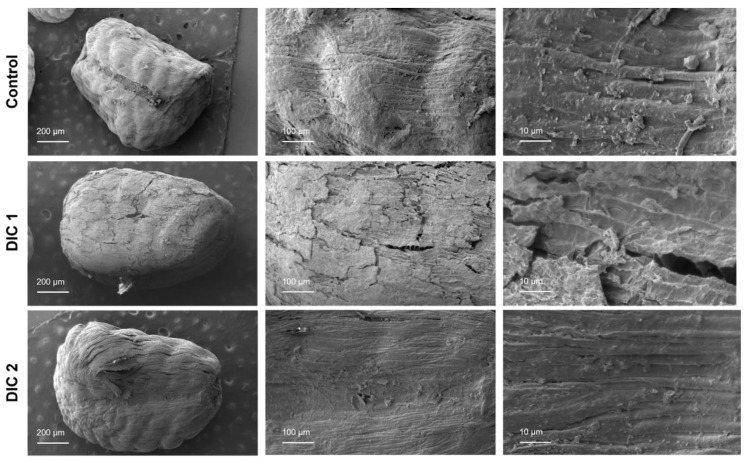
SEM micrographs of cardamom whole seeds treated by DIC and the control (untreated). The bars show three different scales according to the magnitude employed (200 μm for 25× 100 μm for 100×, and 10 μm for 500×).

**Table 1 molecules-28-01093-t001:** Effect of DIC and UAE treatment on cardamom essential oil yield and antioxidant capacity.

	Treatment Conditions		Responses Variables			
Treatment	SSPT ^1^ (°C)	TPT ^2^ (s)	Recovery (g)	Yield % (*w*/*w*)	DPPH % Discoloration	TEAC (uMTE/g)
Control (Only UAE)	NA	NA	0.7799	15.596	57.75	0.60
DIC 1 and UAE	140	30	0.9423	18.846	67.39	0.81
DIC 2 and UAE	165	30	1.0518	21.036	79.48	0.88
DIC 3 and UAE	140	45	1.0941	21.882	71.35	0.94
DIC 4 and UAE	140	30	0.8633	17.266	71.19	0.95
DIC 5 and UAE	158	41	1.1263	22.526	62.44	1.00
DIC 6 and UAE	158	19	0.7139	14.278	71.35	0.75
DIC 7 and UAE	140	30	1.0847	21.694	62.01	1.60
DIC 8 and UAE	122	19	0.9915	19.83	59.42	0.91
DIC 9 and UAE	122	41	0.8099	16.198	70.63	0.90
DIC 10 and UAE	140	30	0.7991	15.982	44.38	2.99
DIC 11 and UAE	115	30	1.0201	20.402	64.50	1.22
DIC 12 and UAE	140	15	0.7877	15.754	57.00	2.01
DIC 13 and UAE	140	30	0.8721	17.442	56.34	2.24

^1^ SSPT = Saturated Steam Processing Temperature; ^2^ TPT = Thermal Processing Time.

**Table 2 molecules-28-01093-t002:** Cardamom essential oil compounds identified by GC-MS.

ID	Compound	CAS Number	Classification
TEOH	α-Terpineol	98-55-5	Monoterpenoid alcohol
OD	2,6-Octadien-1-ol, 3,7-dimethyl-, (Z)-	106-25-2	Monoterpenoid alcohol
GEOH	Geraniol	106-24-1	Monoterpenoid alcohol
LAC	Linalyl acetate	115-95-7	Monoterpenoid
TAC	α-Terpinyl acetate	80-26-2	Menthane Monoterpenoid
HMMC	S-(+)-5-(1-Hydroxy-1-methylethyl)-2-methyl-2-cyclohexen-1-one	60593-11-5	Monoterpenoid
HMAC1	2-((1R,4R)-4-Hydroxy-4-methylcyclohex-2-enyl)propan-2-yl acetate	121958-61-0	Monoterpenoid
HMAC2	2-((1R,4R)-4-Hydroxy-4-methylcyclohex-2-enyl)propan-2-yl acetate	121958-61-0	Monoterpenoid
SAC	Sobrerol 8-acetate	93133-02-9	Terpenoid
DOH	1,6,10-Dodecatrien-3-ol, 3,7,11-trimethyl-, (E)-	40716-66-3	Sesquiterpene alcohol
3-TT	(3E,7E)-4,8,12-Trimethyltrideca-1,3,7,11-tetraene	62235-06-7	Sesquiterpene
DOHT	1,6,10-Dodecatrien-3-ol, 3,7,11-trimethyl-, (E)-	40716-66-3	Farnesane sesquiterpenoid
2-OTAC	2-Oxabicyclo [2.2.2]octan-6-ol, 1,3,3-trimethyl-, acetate	57709-95-2	Terpenoid
HC	exo-2-Hydroxycineole	92999-78-5	Oxane
ACA	8-Acetoxycarvotanacetone	87578-93-6	Menthane monoterpenoid
DOT	2,6,10-Dodecatrien-1-ol, 3,7,11-trimethyl-	4602-84-0	Sesquiterpene alcohol
DAT	2,6,10-Dodecatrienal, 3,7,11-trimethyl-, (E, E)-	502-67-0	Sesquiterpenoid
HEXAD	n-Hexadecanoic acid	57-10-3	Fatty acid
OCAD	9,12-Octadecadienoic acid (Z,Z)-	60-33-3	Polyunsaturated fatty acid
OCAD-E	9-Octadecenoic acid, (E)-	112-79-8	Polyunsaturated fatty acid
OLAD	Oleic Acid	112-80-1	Monounsaturated fatty acid
OA	Octadecanoic acid	57-11-4	Saturated fatty acid
P-HNP	p-Hydroxynorephedrine	552-85-2	Phenylethylamine
Z-DDP	(Z)-3,7-Dimethylocta-2,6-dien-1-yl palmitate	122569-17-9	Saturated fatty acid
GO	Geranyl oleate	81601-03-8	Polyunsaturated fatty acid
PA	Photocitral A	55253-28-6	Monoterpenoid
SOH	ç-Sitosterol	83-47-6	Phytosterols
SIT	ç-Sitostenone	84924-96-9	Ketone

**Table 3 molecules-28-01093-t003:** Experimental design of the DIC pre-treatment parameters.

Sample	Saturated Steam Processing Temperature (°C)	Saturated Steam Processing Time (s)	Saturated Steam Processing Pressure (MPa)
DIC 1	140	30	0.36
DIC 2	165	30	0.70
DIC 3	140	45	0.36
DIC 4	140	30	0.36
DIC 5	158	41	0.58
DIC 6	158	19	0.58
DIC 7	140	30	0.36
DIC 8	122	19	0.21
DIC 9	122	41	0.21
DIC 10	140	30	0.36
DIC 11	115	30	0.17
DIC 12	140	15	0.36
DIC13	140	30	0.36

## Data Availability

Not applicable.

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
