# Peer review of "Antioxidant Activity and GC-MS Profile of Cardamom (Elettaria cardamomum) Essential Oil Obtained by a Combined Extraction Method—Instant Controlled Pressure Drop Technology Coupled with Sonication"

_molecules, 2023, doi:10.3390/molecules28031093_

Round 1

Reviewer 1 Report

In the manuscript, the authors evaluated the effect of coupling the Instant Controlled Pressure Drop (DIC) process to the Ultrasound-Assisted Extraction (UAE) on the cardamom essential oil (CEO) yield, the antioxidant activity of CEO by DPPH scavenging capacity (AOX) and Trolox equivalent antioxidant capacity (TEAC), and the CEO chemical compositions by gas chromatography-mass spectrometry (GC-MS). The work is interesting, and it can be accepted as it is.

Author Response

Dear Reviewer 

Please find attached the file with our responses.

Thank you in advance for your time and support in the revision process

Sincerely

The authors

Reviewer 2 Report

Biological and GC-MS profile of cardamom (Elettaria cardamomum) essential oils obtained by a combined extraction method — Instant Controlled Pressure Drop technology coupled with Sonication.

I read this MS with great interest. This study evaluated the yield and quality of the Instant Controlled Pressure Drop process coupled with Ultrasound-Assisted Extraction of cardamom essential oil (CEO). The antioxidant activity, chemical profile of CEO, and microstructure of seeds were also analyzed.

The results showed that CEO yield significantly increased by DIC (140 °C and 30 s) + UAE compared to the control (22.53% vs. 15.6%). DIC 2 (165 °C, 30 s) showed the highest DPPH inhibition (79.48%) and the best TEAC by the control (only UAE) with 0.60 uMTE/g. The GC/MS analysis showed 28 volatile constituents being the most abundant α-Terpinyl acetate Geranyl oleate, and Oleic Acid. DIC (140 °C and 30 s) + UAE performed the best yield and chemical profile. The SEM microscopy of untreated seeds revealed collapsed structures before the oil cell layer, which reduced extraction yield, contrary to DIC-treated seeds, with more porous structures.  The authors concluded that combining innovative extraction methods could solve the drawbacks of traditional extraction methods.

The MS is interesting, well written and well conducted but there are some recommendations that should be considered to improve the quality of the manuscript. The comments and suggestions are as follows:

-         Line 26-27. The authors used the abbreviations DIC and UAE without definitions. The authors should put each abbreviation in brackets at the first mention and after that they can use the abbreviation only.

-         The abstract should represent all MS parts. The treatments are missed. Despite the authors reported the results in next lines about DIC and DIC2, they are missed as treatments.

-         Line 28. Please define TEAC.

-         Line 145-153. This is confusing. Please revise it carefully. It is well known that if the DPPH values are higher, it means, the concentration of antioxidant compound in the extract is less.

-         Line 456. What the authors mean by (and others were used only…….). Which others? The rest is one lot only.

-         I materials and method chapter, the authors did not clarify how they applied a combined extraction method — Instant Controlled Pressure Drop technology coupled with Sonication. They only explained each of them as a solely treatment. Please provide details step by step about the combined method.

-         Line 570. The conclusion is a little bit long. Please concentrate about your results and recommendations.

Author Response

Dear Reviewer

Please find attached the file in which we revised point by point all your observations.

Thank you in advance for your support in the process

Sincerely, 

The authors

Reviewer 3 Report

Please see Reviewers report attached as separate file. 

Author Response

Dear reviewer

We really appreciate your time and deeper revision of our paper and your providing us with your comments. We have carefully considered them to improve the manuscript. Please find attached the pdf file with the responses to the comments.  Also, we have attached the revised version; all the modifications were highlighted in yellow. We hope after careful revision, it meets your standards. 

Sincerely, 

The authors

Round 2

Reviewer 3 Report

Dear Authors, the revised version of the manuscript was much improved and carefully redrafted. I do appreciate all the effort made by Authors for improving quality of presented work. In my opinion, revised manuscript meets journal requirements and may be suitable for reconsidering its publication.